# Evaluation of the Nigeria national HIV rapid testing algorithm

Nnaemeka C. Iriemenam[1]*, Augustine Mpamugo[2], Akudo Ikpeazu[3], Olumide O. Okunoye[1], Edewede Onokevbagbe[2], Orji O. Bassey[1], Jelpe Tapdiyel[1], Matthias A. Alagi[1], Chidozie Meribe[1], Mukhtar L. Ahmed[1], Gabriel Ikwulono[3], Rose Aguolu[4], Gregory Ashefor[4], Charles Nzelu[3], Akipu Ehoche[2], Babatunde Ezra[2], Christine Obioha[2], Ibrahim Baffa Sule[2], Oluwasanmi Adedokun[2], Nwando Mba[5], Chikwe Ihekweazu[5], Manhattan Charurat[6,7], Brianna Lindsay[7], Kristen A. Stafford[6,7], Dalhatu Ibrahim[1], Mahesh Swaminathan[1], Ernest L. Yufenyuy[8], Bharat S. Parekh[8], Sylvia Adebajo[2], Alash'le Abimiku[6], McPaul I. Okoye[1], for the Evaluation Working Group¶

1 Division of Global HIV and TB, Centers for Disease Control and Prevention, Abuja, Federal Capital Territory, Nigeria, 2 Center for International Health, Education, and Biosecurity, Maryland Global Initiatives Corporation – an affiliate of the University of Maryland, Baltimore, Federal Capital Territory, Nigeria, 3 Federal Ministry of Health, Abuja, Federal Capital Territory, Nigeria, 4 National Agency for the Control of AIDS, Abuja, Federal Capital Territory, Nigeria, 5 National Reference Laboratory, Nigeria Centers for Disease Control, Gaduwa, Federal Capital Territory, Nigeria, 6 Institute of Human Virology, University of Maryland School of Medicine, Baltimore, Maryland, United States of America, 7 Center for International Health, Education, and Biosecurity, Institute of Human Virology, University of Maryland School of Medicine, Baltimore, Maryland, United States of America, 8 International Laboratory Branch, Division of Global HIV and TB, Centers for Disease Control and Prevention, Atlanta, Georgia, United States of America

¶ Membership of the Evaluation Working Group is provided in the Acknowledgments.
* NIriemenam@cdc.gov

**Data Availability Statement:** All relevant data are within the manuscript.

**Funding:** This project was supported by the President's Emergency Plan for AIDS Relief

## Abstract

Human Immunodeficiency Virus (HIV) diagnosis remains the gateway to HIV care and treatment. However, due to changes in HIV prevalence and testing coverage across different geopolitical zones, it is crucial to evaluate the national HIV testing algorithm as false positivity due to low prevalence could be detrimental to both the client and the service delivery. Therefore, we evaluated the performance of the national HIV rapid testing algorithm using specimens collected from multiple HIV testing services (HTS) sites and compared the results from different HIV prevalence levels across the six geopolitical zones of Nigeria. The evaluation employed a dual approach, retrospective, and prospective. The retrospective evaluation focused on a desktop review of program data (n = 492,880) collated from patients attending routine HTS from six geopolitical zones of Nigeria between January 2017 and December 2019. The prospective component utilized samples (n = 2,895) collected from the field at the HTS and tested using the current national serial HIV rapid testing algorithm. These samples were transported to the National Reference Laboratory (NRL), Abuja, and were re-tested using the national HIV rapid testing algorithm and HIV-1/2 supplementary assays (Geenius to confirm positives and resolve discordance and multiplex assay). The retrospective component of the study revealed that the overall proportion of HIV positives, based on the selected areas, was 5.7% (28,319/492,880) within the study period, and the discordant rate between tests 1 and 2 was 1.1%. The prospective component of the study indicated no significant differences between the test performed at the field using the national

(PEPFAR) through the U.S. Centers for Disease Control and Prevention (CDC) under the Grant Number NU2GGHOO2108 NOA 02-03 to the University of Maryland, Baltimore. The funders had no role in study design, data collection and analysis, decision to publish, or preparation of the manuscript. Disclaimer The findings and conclusions in this report are those of the authors and do not necessarily represent the views of the U.S. Centers for Disease Control and Prevention. The use of trade names is for identification purposes only and does not constitute an endorsement by the U.S. Centers for Disease Control and Prevention or the U.S. Department of Health and Human Services.

**Competing interests:** The authors have declared that no competing interests exist.

HIV rapid testing algorithm and the re-testing performed at the NRL. The comparison between the test performed at the field using the national HIV rapid testing algorithm and Geenius HIV-1/2 supplementary assay showed an agreement rate of 95.2%, while that of the NRL was 99.3%. In addition, the comparison of the field results with HIV multiplex assay indicated a sensitivity of 96.6%, the specificity of 98.2%, PPV of 97.0%, and Kappa Statistic of 0.95, and that of the NRL with HIV multiplex assay was 99.2%, 99.4%, 99.0%, and 0.99, respectively. Results show that the Nigeria national serial HIV rapid testing algorithm performed very well across the target settings. However, the algorithm's performance in the field was lower than the performance outcomes under a controlled environment in the NRL. There is a need to target testers in the field for routine continuous quality improvement implementation, including refresher trainings as necessary.

## Introduction

Human Immunodeficiency Virus (HIV) diagnosis is the critical entry point to accessing HIV care and treatment, especially for those diagnosed as HIV positive. HIV diagnosis also facilitates access to preventive services to those testing HIV negative based on the national HIV testing algorithm. It is the first pillar of the Joint United Nations Program on HIV/AIDS (UNAIDS) 95-95-95 cascade [1]. The national HIV rapid testing algorithm is achieved using the right testing strategy that incorporates two or more appropriate tests in a serial or parallel algorithm [2]. Therefore, developing and deploying a suitable HIV testing algorithm is critical in ensuring a country's quality of HIV testing for HIV diagnostics and screening programs.

Nigeria adopted a two-test national HIV rapid testing algorithm [3] in 2007 after phases 1 and 2 evaluations [4]. The evaluation report recommended a serial testing algorithm in three combinations: 1) Determine, Stat-Pak, and Bundi; 2) Unigold, Stat-Pak, and Bundi; and 3) Determine, Unigold, and Stat-Pak. As a result, the Government of Nigeria adopted the third option as a serial algorithm comprising of the Alere Determine HIV rapid test as the first test (T1), Unigold HIV rapid test as the second test (T2), and Stat-Pak HIV rapid test (T3) as the tie-breaker [3]. HIV-positive diagnosis requires any two reactive tests combination, T1/T2 or T1/T3; the T3 tie-breaker test is used only when the T1 and T2 results are discordant.

The adopted national HIV/AIDS rapid testing algorithm in use for the national HIV program was adopted for the 2018 Nigeria HIV/AIDS Indicators and Impact Survey (NAIIS), a population based survey that estimated the national HIV prevalence, incidence, and viral load suppression [5]. The survey showed a much lower national HIV prevalence status of 1.4% in Nigeria compared to the 4.1% from previous surveys [6, 7]. However, analysis of the NAIIS data indicated that the concordance rate between Test 1 and Test 2 was low (56.5%), and the positive predictive value (PPV) of HIV-positive diagnosis was 94.5% when confirmed with Geenius supplementary assay [8]. The World Health Organization (WHO), using mathematical modeling, recommends three consecutive test algorithm requiring all three reactive tests if the national HIV prevalence is <5% due to anticipated low PPV [2]. The WHO recommendation also indicated that the national HIV rapid testing algorithm should have at least a PPV of 99% and a combination of HIV tests with a sensitivity of ≥99% and specificity of ≥98% [2]. This further necessitated the need to reevaluate the existing national HIV rapid testing algorithm to reexamine its performance across different States with varying HIV prevalence levels in Nigeria, now that NAIIS has shown an overall low HIV national prevalence of 1.4%.

This study aims to evaluate the performance of the national HIV rapid testing algorithm at different HIV prevalence levels across the six geopolitical zones of Nigeria. The study utilized a two-fold approach. Firstly, a retrospective evaluation of the performance of the current national HIV rapid testing algorithm using program data collected from January 2017 to December 2019 from routine HIV testing sites across the six geopolitical zones of Nigeria. Secondly, a prospective sample collection to assess the PPV and negative predictive value (NPV) of the current national HIV rapid testing algorithm in States with different HIV prevalence levels.

## Materials and methods

The study was a cross-sectional evaluation implemented in two phases. First was the retrospective analysis of the routine HIV testing program data to establish historical discordance rates from routine program implementation by geopolitical zones, State, and testing points. Secondly, a prospective analysis of samples collected from HIV testing sites across the six geopolitical zones with varied HIV prevalence. Two States per geopolitical zone and two sites per State with an annual minimum test capacity of 1,000 HIV rapid tests per year were purposively selected for the retrospective component of the evaluation. For the prospective component of the study, one State per geopolitical zone and three sites per State were purposively selected based on considerations for ease of travel logistics and an annual testing capacity of 1,000 HIV tests.

## Retrospective study

In total, twelve States from the six geopolitical zones (two States per zone and two sites per State) with varied HIV prevalence based on the NAIIS report [5] were selected. The sites were purposively selected from the HIV testing sites of the six geopolitical zones of Nigeria. The two States per geopolitical zone were selected based on the NAIIS 2018 HIV prevalence and classified into low (0.1%-1.0%), medium (1.1%-3.0%), and high (>3.0%) categories considering the national program priorities (Table 1).

First, the study team abstracted routine service data on patients' age, sex, testing point, and HIV rapid test results based on the national serial HIV rapid testing algorithm (Fig 1) without personal identifiers from the paper-based National Daily HIV Testing Register from January 2017 to December 2019. The abstracted data were then entered into an electronic custom-built data collection tool using Android Studio for analysis.

## Prospective study

The second phase of this study was a prospective evaluation of the performance of the national HIV rapid testing algorithm. One State per geopolitical zone based on NAIIS 2018 results was selected for the prospective study. The selection was based on States with low, medium, and high levels of HIV prevalence (Table 1). Three HIV testing sites per State were selected, making a total of 18 purposively selected sites based on the ease of logistics and an annual testing capacity of 1,000 HIV tests. In addition, the selection of the sites was purposeful based on the following criteria: testing sites with the availability of HIV Testing Services (HTS); testing sites with low, medium, or high yield of HIV positive cases; testing sites with ease of access to transportation and sample shipment to the National Reference Laboratory (NRL), Gaduwa; as well as sites that are not in conflict-prone areas. Clients aged 15–64 years seeking HTS services at the selected health facilities sites and consented to participate were consecutively chosen for the study. Clients who declined to participate were excluded from the study but had their HTS based on the national HIV counseling and testing guidelines [3].

**Table 1. The number of samples collected by States, geographical zones, HIV prevalence, and testing sites.**

| Selected States | Samples collected from the State (N) | Geographical Zones | HIV Prevalence (NAIIS) | Prevalence Category | Testing Sites (N) |
|---|---|---|---|---|---|
| **Retrospective study (n = 492,880)** | | | | | |
| Katsina | 23,781 | North West | 0.3 | Low | 2 |
| Ekiti | 43,678 | South West | 0.8 | Low | 2 |
| Ebonyi | 36,521 | South East | 0.8 | Low | 2 |
| Kaduna | 32,350 | North West | 1.1 | Medium | 2 |
| Gombe | 19,705 | North East | 1.3 | Medium | 2 |
| FCT | 98,230 | North Central | 1.6 | Medium | 2 |
| Ogun | 58,793 | South West | 1.6 | Medium | 2 |
| Delta | 27,629 | South South | 1.9 | Medium | 2 |
| Enugu | 21,365 | South East | 2.0 | Medium | 2 |
| Taraba | 71,678 | North East | 2.9 | Medium | 2 |
| Rivers | 27,064 | South South | 3.8 | High | 2 |
| Benue | 32,086 | North Central | 5.3 | High | 2 |
| **Prospective study (n = 2,895)** | | | | | |
| Katsina | 443 | North West | 0.3 | Low | 3 |
| Ekiti | 457 | South West | 0.8 | Low | 3 |
| Gombe | 499 | North West | 1.3 | Medium | 3 |
| Enugu | 476 | South East | 2.0 | Medium | 3 |
| Rivers | 520 | South Ssouth | 3.8 | High | 3 |
| Benue | 500 | North Central | 5.3 | High | 3 |

NB: N = number, FCT = Federal Capital Territory.

Blood samples were collected from the consenting participants who met the eligibility criteria. Written informed consents/assents were duly requested from all potential study participants, and consents were also obtained to have their remnant specimens stored for future research. In addition, informed consent forms were administered to adults (18–64 years), while informed parental consent forms were administered to parents/guardians of study participants aged 15–17 years. The participants 15–17 years were required to sign the assent form. The WHO guideline recommended consecutive sampling to obtain a minimum of 200 HIV-positive and 200 HIV-negative samples [9]. We proposed a sample size estimate of 200 HIV-positive and 300 HIV-negative samples from each geopolitical zone. Consenting study participants had their samples collected in a stepwise order: A finger prick was used to collect blood specimens for the first test at the HIV point of service testing (POST) following the standard protocol. The HIV POST personnel at the facility performed HIV rapid tests according to the national serial HIV rapid testing algorithm (Fig 1) [3]. If test 1 (Determine) was non-reactive to HIV (HIV-negative result), the algorithm testing was concluded and reported at the POST, but the study participant was escorted to the Site testing laboratory for a venous blood collection. A 10 ml EDTA vacutainer tube was used to collect venous blood from the client following standard protocol. This process was done consecutively on all the confirmed HIV-negatives until the 300 negative specimens were collected from each geopolitical zone. If test 1 (Determine) was reactive, the participant was escorted to the laboratory for venous sample collection. The testing process was completed in the laboratory using the venous blood sample by a trained laboratory scientist. A 10 ml EDTA vacutainer tube was used to collect venous blood from the client. For participants that were reactive on Determine, the laboratory scientist used the EDTA blood sample to perform the second test (Unigold), and if non-reactive, a third test

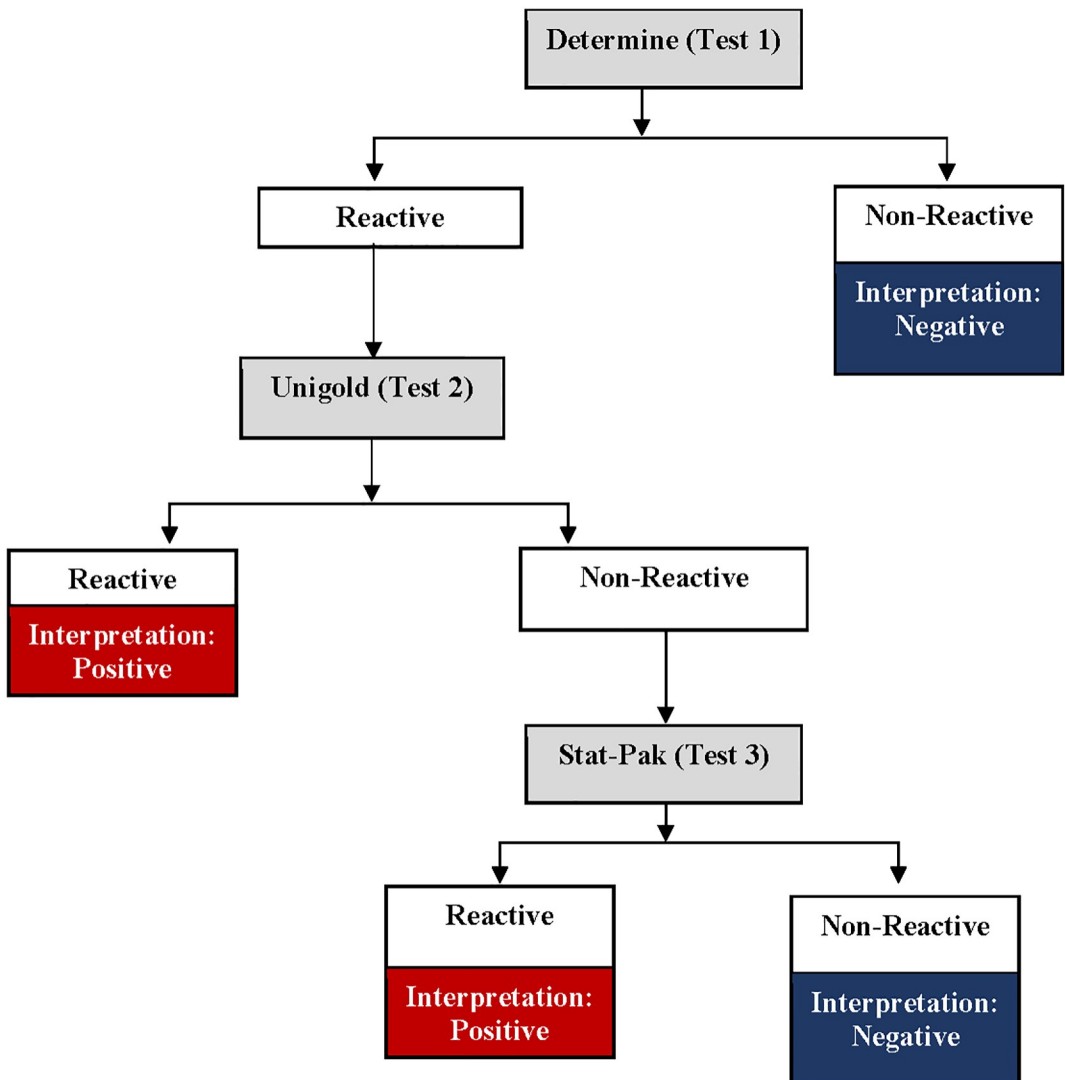

**Fig 1. Nigeria serial HIV rapid testing algorithm.**

(Stat-Pak) as tie-breaker was conducted, in line with the national serial HIV rapid testing algorithm [3]. This was done on all confirmed HIV-positives based on the national serial HIV rapid testing algorithm until the required number of positive specimens (target 200) were consecutively collected from the geopolitical zone. This sample collection approach was adopted to minimize the discomfort of multiple finger pricks and venous punctures for the study participants. Sampling was done until the required sample sizes for HIV-positive, and HIV-negative samples were collected from each of the six geopolitical zones. In total, 1200 HIV-positive and 1800 HIV-negative specimens were targeted from all the zones following the WHO guidelines for appropriate evaluations of HIV testing technologies in Africa [9]. However, at the time data collection was discontinued, 1,078 (90%) HIV-positive and 1,817 (101%) HIV-negative specimens had been collected from all the geopolitical zones. Sample collection was stopped at the end of the funding stream for the project.

All field results were documented and blinded until analysis. Plasma was separated from the collected samples by centrifugation (1500 xg for 10 minutes or ~3000 rpm). The plasma

was aspirated using sterile disposable plastic pasteur pipettes, divided into two aliquots of about 1.5 mL volume in cryo-tubes, and labeled with pre-printed labels containing de-identified sample identification codes. Each participant identification number was an alpha-numeric code that defined the State, the facility, and a randomly distributed numeric value. The aliquots were stored at a designated central collation center in the States at -20˚C freezers till transported in -20˚C Crēdo Cube [10] to the NRL.

At the NRL, under controlled conditions (temperature-regulated laboratory between 22˚C-26˚C), the laboratory scientists re-tested all the 2,895 specimens using the national HIV rapid testing algorithm. The NRL laboratory staff received a refresher training and competency assessment just prior to the evaluation of the national HIV rapid testing algorithm. The training consisted of didactic re-orientation of the NRL laboratory staff on HIV serology, the national HIV rapid testing algorithm, quality assurance, sample transportation and storage, and competency assessment of the staff on the respective assays. Any laboratory staff who did not score 100% on the practical competency assessment was dropped from the testing. Additionally, all samples with HIV-positive test results based on the NRL testing were re-tested using a specific HIV supplementary assay (Bio-Rad Geenius HIV-1/2 Supplemental Assay) [11] as a confirmatory assay. Samples with discordant results between Test 1 (Determine HIV-1/2) and Test 2 (Uni-Gold HIV-1/2) at the NRL, and discordant results between the field and NRL testing were also re-tested using Geenius supplementary assay to resolve discordance.

Furthermore, an HIV multiplex assay for concurrent HIV diagnosis and serotyping of HIV-1 and HIV-2 was performed on all the 2,895 specimens to compare the performance of the national HIV rapid testing algorithm. The coupling procedure for HIV antigens to beads and the selected serotyping antigens had been previously described [12]. Briefly, the coupling of beads for HIV diagnosis and HIV-2 serotyping was done using HIV-1 p24-gp41 fusion protein and HIV-2 peptide from the gp36 immunodominant region, respectively. The assay conditions, including the concentration of the coupled antigens, were optimized using well-characterized specimens with known HIV status and HIV-2 specimens. HIV multiplex testing was done in duplicate by two different laboratory scientists. Any discordance, defined as a percentage difference of >20% between two testers or two testers with different classifications (positive and negative), was repeated in triplicate, and the median was used. Plates were read using the Luminex xMAP MAGPIX System (Luminex Corporation, Austin, USA), including the HIV negatives from the national HIV rapid testing algorithm.

## Ethical clearance

The study received ethical clearance from the National Health Research Ethics of Nigeria Committee (NHREC), the UMB Institutional Review Board, and the United States Centers for Disease Control and Prevention (CDC). This activity was reviewed by CDC and was conducted consistent with applicable federal law and CDC policy with the Code of Federal Regulations 45 CFR § 46.116; 21 CFR § 50.25(a)(b).

## Statistical analysis

Proportions were used to summarize qualitative variables and cross-tabulations using 2x2 contingency tables. Demographic characteristics, age, sex, State, and testing points were analyzed for the retrospective study, while State, HIV prevalence category, and testing points were analyzed for the prospective study. PPV, NPV, Kappa statistics, and agreement rates, with the corresponding 95% confidence interval (CI), were computed for the prospective study, and differences in proportion between field HIV status results and the re-testing performed at the NRL were compared using McNemar's test. Agreement rate and Kappa statistics were

determined to estimate inter-rater reliability between field and NRL results as well as against the reference testing results, using Geenius assay as the HIV confirmatory test and HIV multiplex bead assay as a complementary supplementary test. Data from the field testing sites and NRL were analyzed using SAS 9.4 (SAS Institute, Cary, NC, USA) and Microsoft Excel (Microsoft Corporations, Redmond, USA).

## Results

### Retrospective evaluation

Fig 2 shows the national HIV rapid testing algorithm and the number of abstracted results from the HTS sites. Out of the 492,880 test results, 94.2% (464,331) were non-reactive and 5.8% (28,550) were reactive on Test 1 (Determine). Of those that were reactive on Test 1,

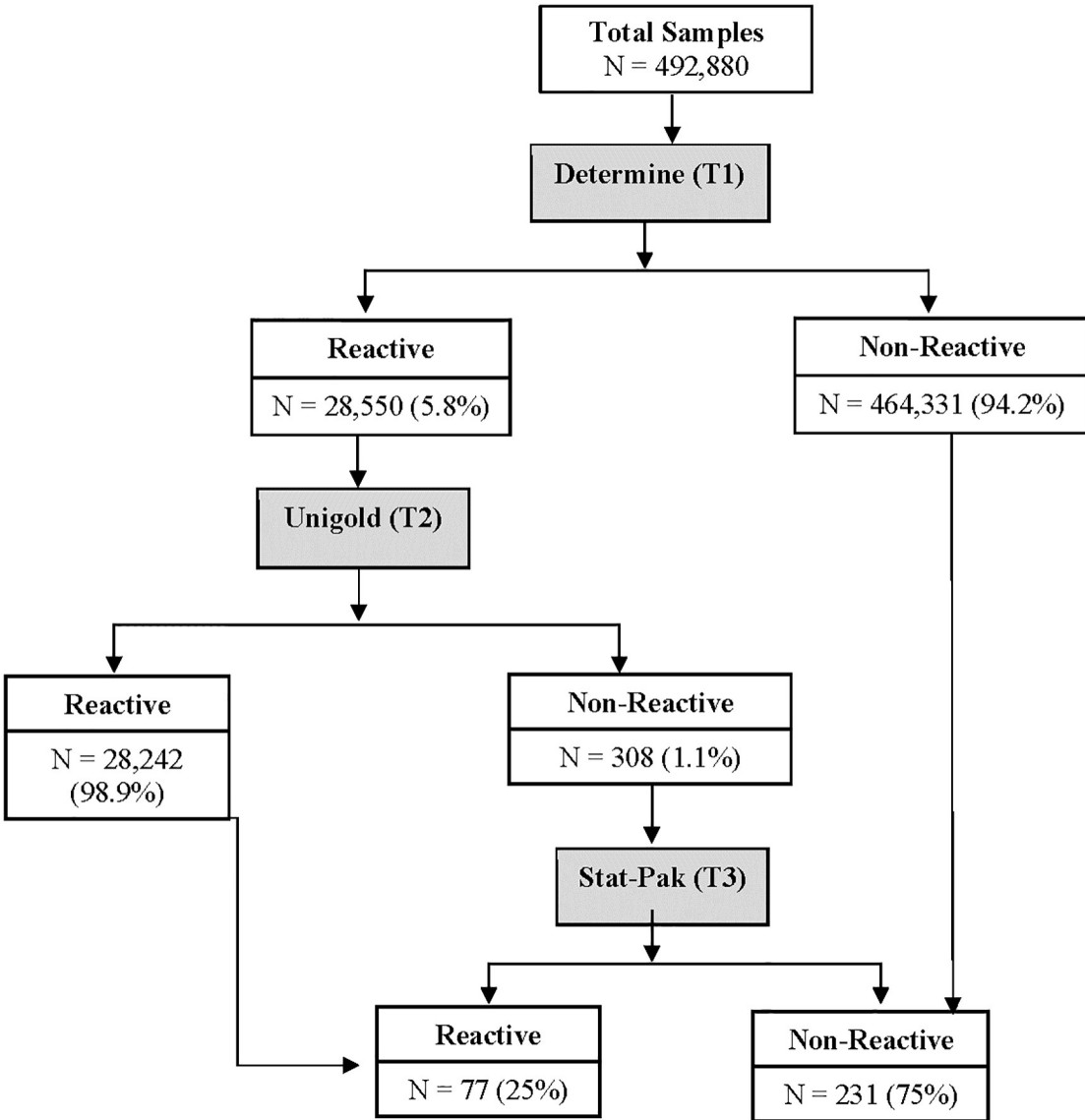

**Fig 2. Flowchart of the national serial HIV rapid testing algorithm performed at the testing sites for retrospective evaluation.** N = number, T = test.

**Table 2. Retrospective analysis of the discordance rates across age, states, and testing points.**

| Characteristics | Total (N = 492,880) | Determine | | Unigold | | | Discordant Rates |
|---|---|---|---|---|---|---|---|
| | | Total Reactive | | Total Reactive | | Positive Concordant rates | |
| | | (N = 28,550) | | (N = 28,242) | | | |
| | | N | % | N | % | % | % |
| **Age** | | | | | | | |
| Adult | 478,168 | 28,221 | 98.9 | 27,914 | 98.8 | 98.9 | 1.1 |
| Pediatric | 14,712 | 329 | 1.2 | 328 | 1.2 | 99.7 | 0.3 |
| **State** | | | | | | | |
| Benue | 32,086 | 1,795 | 6.3 | 1,783 | 6.3 | 99.3 | 0.7 |
| FCT | 98,230 | 2,498 | 8.8 | 2,494 | 8.8 | 99.8 | 0.2 |
| Gombe | 19,705 | 1,400 | 4.9 | 1,397 | 4.9 | 99.9 | 0.1 |
| Taraba | 71,678 | 4,170 | 14.6 | 4,116 | 14.6 | 98.7 | 1.3 |
| Kaduna | 32,350 | 2,184 | 7.7 | 2,174 | 7.7 | 99.5 | 0.5 |
| Katsina | 23,781 | 1,806 | 6.3 | 1,806 | 6.4 | 100 | 0 |
| Ebonyi | 36,521 | 1,603 | 5.6 | 1,528 | 5.4 | 95.3 | 4.7 |
| Enugu | 21,365 | 3,159 | 11.1 | 3,159 | 11.2 | 100 | 0 |
| Delta | 27,629 | 3,233 | 11.3 | 3,138 | 11.1 | 97.1 | 2.9 |
| Rivers | 27,064 | 2,769 | 9.7 | 2,765 | 9.8 | 99.9 | 0.1 |
| Ekiti | 43,678 | 1,129 | 3.9 | 1,113 | 3.9 | 98.6 | 1.4 |
| Ogun | 58,793 | 2,804 | 9.8 | 2,769 | 9.8 | 98.8 | 1.2 |
| **Testing points** | | | | | | | |
| FP | 4,851 | 180 | 0.6 | 180 | 0.6 | 100 | 0 |
| *Standalone | 214 | 15 | 0.05 | 15 | 0.1 | 100 | 0 |
| STI | 281 | 49 | 0.2 | 49 | 0.2 | 100 | 0 |
| Ward | 4,710 | 210 | 0.4 | 207 | 0.4 | 98.6 | 1.4 |
| **Others | 80,769 | 2,274 | 8.0 | 2,270 | 8 | 99.8 | 0.2 |
| Outreach | 23,988 | 248 | 0.9 | 247 | 0.9 | 99.6 | 0.4 |
| HTS | 218,915 | 15,606 | 54.7 | 15,485 | 54.8 | 99.2 | 0.8 |
| OPD | 56,376 | 3,585 | 12.6 | 3,524 | 12.5 | 98.3 | 1.7 |
| Lab | 94,905 | 6,336 | 22.2 | 6,221 | 22 | 98.2 | 1.8 |
| †Pediatric Services | 7,871 | 47 | 0.2 | 44 | 0.2 | 93.6 | 6.4 |

NB: N = numbers, FP = family planning, STI = sexually transmitted infection clinic, OPD = outpatient department.

*Standalone are HTS sites providing voluntary counseling and testing mainly on a client-initiated testing basis.

**Others include TB unit and Key Populations.

† Refers to the pediatric outpatient department, well-child clinics, immunization clinics, and nutrition clinics.

98.9% (28,242) were reactive on Test 2, while 1.1% (308) were non-reactive on Test 2 (Unigold). Of the non-reactive results with Test 2, the results show that 75% (231) were non-reactive and 25% (77) were reactive using the tie-breaker (Test 3 –Stat-Pak).

The overall discordant rate between Test 1 and Test 2 of the national HIV rapid algorithm was 1.1%. The concordant and discordant rates across age, States, and testing points are shown in Table 2. The highest discordant rate was observed in Ebonyi (4.7%), followed by Delta (2.9%), and none observed in Katsina (0%) and Enugu (0%), respectively. When categorized by testing points, pediatric services showed the highest discordant rate (6.4%), possibly due to a very low number of samples tested through the testing points (n = 47), followed by inpatient services (2.8%, n = 107).

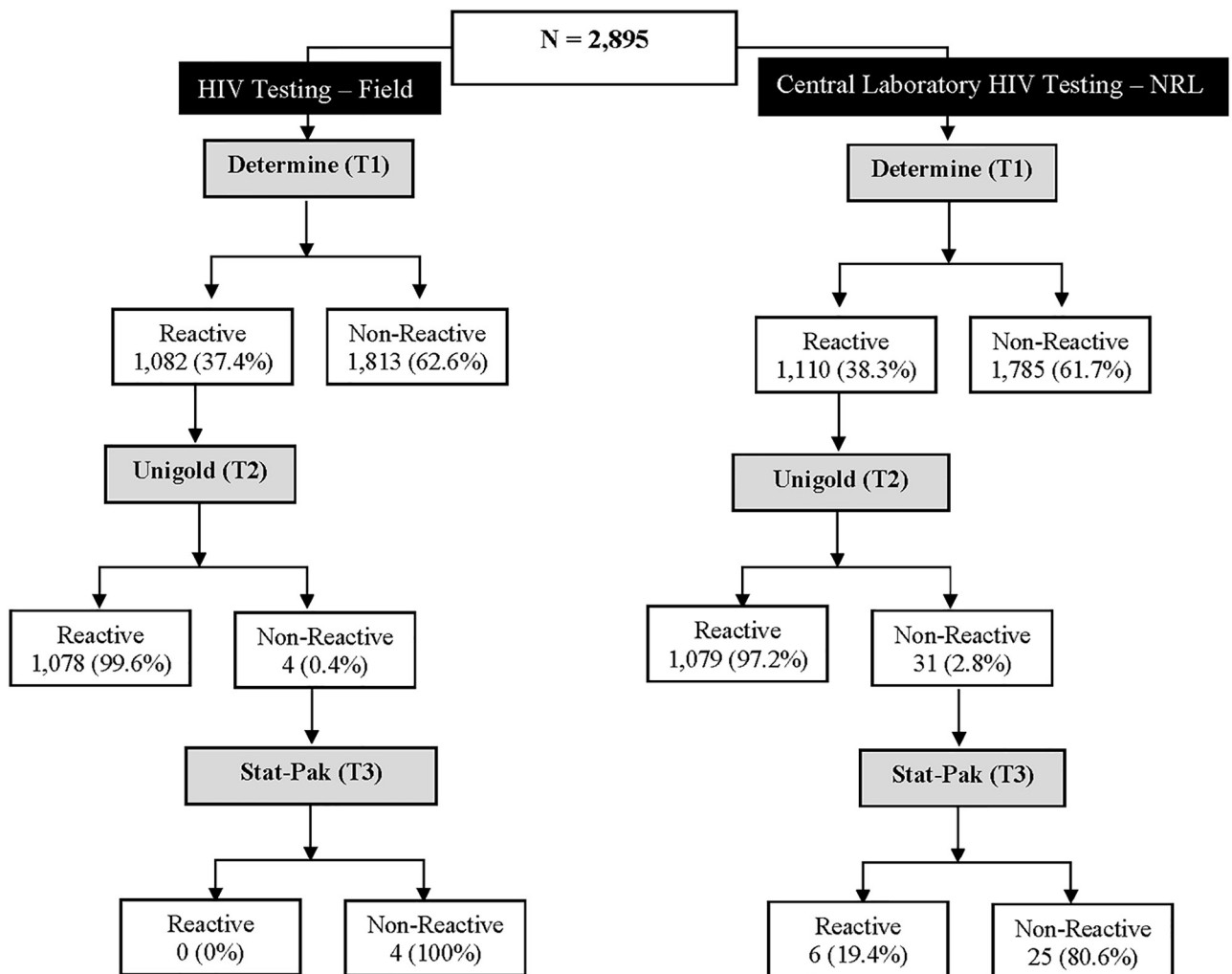

**Fig 3. Flowchart of the national serial HIV rapid testing algorithm performed in the field and the NRL for the prospective evaluation.**

### Prospective evaluation

In total, 2,895 samples were collected for the prospective evaluation. Fig 3 shows the flowchart of the national serial HIV rapid testing algorithm performed in the field at the testing sites and then repeated at the central laboratory (NRL). At the field, 62.6% of those tested were non-reactive, and 37.4% were reactive with Test 1. Similarly, 61.7% of the re-testing at the NRL were non-reactive, and 38.3% were reactive. However, the discordant rate between Tests 1 and 2 at the field and NRL were 0.4% and 2.8%, respectively.

Table 3 shows the 2x2 summary of the national Serial HIV rapid testing algorithm performed at the field and the re-testing performed at the NRL.

The overall percentage agreement rate was 98.2%, 95% CI [96.2, 100.0]. The performance comparison of the national serial HIV rapid testing algorithm between the field and NRL across States, prevalence categories, and testing points is shown in Table 4.

The overall performance indicates a Kappa statistic of 0.96, 95% CI [0.93, 0.99], which revealed no statistically significant difference between the tests performed at the field and the NRL (P = 0.327). When stratified by geopolitical zones (States), prevalence category, and

**Table 3. Summary of the national serial HIV rapid testing algorithm performed in the field compared with the re-testing at the NRL from the prospective study.**

| | | National HIV Rapid Testing Algorithm (NRL) | | |
|---|---|---|---|---|
| | | Positive N (%) | Negative N (%) | Total |
| National HIV Rapid Testing Algorithm (Field) | Positive | 1,056 (97.3%) | 22 (1.2%) | 1,078 |
| | Negative | 29 (2.7%) | 1,788 (98.8%) | 1,817 |
| | Total | 1,085 | 1,810 | 2,895 |

testing points, there were no statistical differences with the PPVs, NPVs, Kappa statistics, and agreement rates across the categories.

To further investigate the status of samples with positive or discordant results, all samples with HIV positive results, all samples with discordant results between Tests 1 and 2, and all samples that showed discordant results between the field and NRL testing were subjected to the Geenius HIV-1/2 supplementary assay. S1 Table shows the comparison of field and NRL final results with Geenius HIV-1/2 supplementary assay. The result indicates a lower agreement rate 95.2%, 95% CI [93.2, 97.2] between the field results and Geenius HIV-1/2 supplementary assay (S1 Table). In addition, the comparison of the national serial HIV rapid testing algorithm performed at the NRL with Geenius HIV-1/2 supplementary assay showed a stronger agreement rate of 99.3%, 95% CI [97.3, 100.0] (S1 Table).

To further analyze the performance of the national HIV rapid testing algorithm, all samples were re-tested using the HIV multiplex assay, which is a newly developed laboratory-based test

**Table 4. Performance comparison of the national serial HIV rapid testing algorithm between the field and NRL across states, prevalence categories, and testing points using NRL re-testing results as a reference.**

| Overall Performance | PPV % (95% CI) | NPV % (95% CI) | Kappa % (95% CI) | Agreement rate % (95% CI) | McNemar's Test (P value) |
|---|---|---|---|---|---|
| | 98.0% (97.1–98.8) | 98.4% (97.8–98.9) | 0.96 (0.93–0.99) | 98.2% (96.2–100.0) | P = 0.327 |
| **State** | | | | | |
| Katsina (NW) | 98.8% (98.7–98.8) | 96.0% (95.9–96.0) | 0.94 (0.91–0.97) | 97.1% (95.1–99.1) | P = 0.0013 |
| Gombe (NE) | 98.0% (98.5–100.0) | 98.0% (97.9–98.0) | 0.96 (0.93–0.98) | 98.0% (96.0–99.9) | P = 0.5271 |
| Enugu (SE) | 97.3% (95.3–99.3) | 98.6% (97.0–100.0) | 0.96 (0.93–0.99) | 98.1% (96.1–100.0) | P = 0.7389 |
| Benue (NC) | 96.1% (94.1–98.1) | 98.6% (97.0–100.0) | 0.95 (0.92–0.98) | 97.6% (95.6–99.6) | P = 0.2482 |
| Ekiti (SW) | 99.1% (97.1–100.0) | 99.4% (97.4–100.0) | 0.98 (0.96–1.00) | 99.3% (97.3–100.0) | P = 0.5637 |
| Rivers (SS) | 99.0% (97.0–100.0) | 99.4% (97.4–100.0) | 0.98 (0.97–1.00) | 99.2% (97.2–100.0) | P = 1.0000 |
| **Prevalence category** | | | | | |
| Low | 98.9% (96.9–100.0) | 97.9% (95.9–99.9) | 0.96 (0.94–0.98) | 98.2% (96.2–100.0) | P = 0.0124 |
| Middle | 97.7% (95.7–99.7) | 98.3% (96.3–100.0) | 0.96 (0.94–0.98) | 98.1% (96.1–100.0) | P = 0.8185 |
| High | 97.5% (95.5–99.5) | 99.0% (97.0–100.0) | 0.97 (0.95–0.98) | 98.4% (96.4–100.0) | P = 0.3173 |
| **Testing points** | | | | | |
| OPD (n = 372) | 98.3% (96.3–100.0) | 99.0% (97.0–100.0) | 0.96 (0.12–1.00) | 98.9% (94.7–98.7) | P = 1.0000 |
| Standalone (n = 49) | 100.0% (98.0–100.0) | 95.7% (93.7–97.7) | 0.96 (0.88–1.00) | 98.0% (96.0–99.9) | P = 0.3173 |
| HTS (n = 1,733) | 97.8% (95.8–99.8) | 98.3% (96.3–100.0) | 0.96 (0.12–1.00) | 98.1% (96.1–100.0) | P = 1.0000 |
| Laboratory (n = 459) | 97.9% (95.9–99.7) | 99.7% (97.7–100.0) | 0.98 (0.96–1.00) | 99.1% (97.1–100.0) | P = 0.3173 |
| Ward* (n = 103) | 93.8% (91.3–95.3) | 96.6% (94.5–98.5) | 0.85 (0.72–0.99) | 96.1% (94.0–98.0) | P = 0.3173 |
| Others** (n = 179) | 100% (98.0–100.0) | 96.0% (94.0–98.0) | 0.91 (0.80–1.00) | 94.3% (89.5–99.2) | P = 0.1797 |

Note: PPV = positive predictive value, NPV = negative predictive value, CI = confidence interval, NW = north west, NE = north east, SE = south east, NC = north central, SW = south west, SS = south south, n = number.

*Ward (Inpatient Services),

**Others (Tuberculosis unit (TB), family planning (FP), sexually transmitted infection (STI), Outreach, Key populations, Pediatric Services).

**Table 5. The comparison of the national serial HIV rapid testing algorithm performed at the field and NRL with HIV multiplex assays from the prospective study.**

| | Sensitivity % (95% CI) | Specificity % (95% CI) | HIV Multiplex Assay (n = 2,895) | | | |
|---|---|---|---|---|---|---|
| | | | PPV % (95% CI) | NPV % (95% CI) | Agreement rate % (95% CI) | Kappa Statistics % (95% CI) |
| National HIV Rapid Testing Algorithm (Field) | 96.6 (95.5–97.7) | 98.2 (97.6–98.8) | 97.0 (95.5–97.7) | 98.0 (97.6–98.8) | 97.6 (95.6–99.6) | 0.95 (0.93–0.97) |
| National HIV Rapid Testing Algorithm (NRL) | 99.2 (98.4–99.6) | 99.4 (99.0–99.8) | 99.0 (98.9–99.9) | 99.5 (99.0–99.8) | 99.3 (97.3–101.3) | 0.99 (0.97–1.00) |

whose performance was evaluated using NAIIS survey specimens. The assay performed well with high sensitivity and specificity (99.7% and 99.4%, respectively) [13]. All the 2,895 specimens were tested using the HIV multiplex assay. The field and HIV multiplex assay results comparison showed a lower sensitivity of 96.6%, specificity of 98.2%, PPV of 97.0%, NPV of 98.0%, and Kappa statistics of 0.95, 95% CI [0.93, 0.97] [Table 5] of the multiplex results. However, the NRL comparison with the HIV multiplex assay showed a higher sensitivity of 99.2%, specificity of 99.4%, PPV of 99.0%, NPV of 99.5%, and Kappa statistics of 0.99, 95% CI [0.97, 1.00] [Table 5]. In addition, the false-positive rate was higher with the field results than the NRL results using HIV multiplex assay as the reference (1.8% and 0.6%) [Table 6]. Also, the false-negative rate was higher in the field than the NRL results when HIV multiplex assay (3.4% and 0.8%) was used as the reference test.

## Discussion

The evaluation of the national HIV rapid testing algorithm performed at the field showed lower sensitivity and PPV compared to re-testing performed at the NRL and the test results from HIV supplementary assay Multiplex. In addition, the laboratory-based testing appears consistent with the WHO recommended performance expectations for national HIV testing algorithms and met the thresholds for sensitivity ($\geq$ 99%), specificity ($\geq$ 98%), and PPV ($\geq$ 99%). Furthermore, agreement rates were consistently above 94%, and Kappa statistics were above 0.85 for all analyses, thus suggesting good inter-rater concurrence between field and NRL results, field and supplementary assays (Geenius and Multiplex), as well as NRL and supplementary assays. However, agreement was better between NRL and the supplementary assays compared to field and the complementary assays, thus strengthening the finding of a better performance of the national serial HIV testing algorithm at the NRL compared to the field.

The outcome of this study also showed a low discordant rate of 1.1% between Tests 1 and 2, from the retrospective study that included testing data from 492,880 individual HTS clients in the field. In comparison, the prospective study showed discordant rates ranging from 0.4% to 2.8% in the field and laboratory, all of which were significantly lower than the discordant rate

**Table 6. Summary of the national serial HIV rapid testing algorithm compared with HIV multiplex assay results.**

| | | HIV Multiplex Assay | | |
|---|---|---|---|---|
| | | Positive N (%) | Negative N (%) | Total |
| National HIV Rapid Testing Algorithm (Field) | Positive | 1,046 (96.6%) | 32 (1.8%) | 1,078 |
| | Negative | 37 (3.4%) | 1,780 (98.2%) | 1,817 |
| | Total | 1,083 | 1,812 | 2,895 |
| National HIV Rapid Testing Algorithm (NRL) | Positive | 1,074 (99.2%) | 11 (0.6%) | 1,085 |
| | Negative | 9 (0.8) | 1,801 (99.4%) | 1,810 |
| | Total | 1,083 | 1,812 | 2,895 |

of 43.4% observed in the NAIIS [8], as well as below the allowable discordance threshold of less than 5% recommended by WHO [2]. Thus suggesting that the observations from the NAIIS were specific to the survey and not representative of the field performance of the algorithm. WHO recommends that the discordant rate between the first and second tests of the national HIV rapid testing algorithm should not exceed 5% [14], though the discordant rate between Test 1 and Test 2 may be a function of HIV prevalence. However, there was no association with HIV prevalence and discordant rate across the HIV prevalence categories, States, and testing points in this study. The performance of the current national HIV rapid testing algorithm, with regards to sensitivity, specificity, PPV, and NPV, when performed in the laboratory by laboratory professionals, meets the WHO recommended performance characteristics for national HIV testing algorithms. However, these performance characteristics were found to be lower when performed by regular testers in the field, outside a controlled laboratory environment. Given that the field and out in the community is where the majority of HIV testing services are delivered, the outcome of this study highlights the urgent need for routine and consistent implementation of quality assurance and continuous quality improvement processes, including refresher trainings for testers in the field.

A multicentre study of the performance of the HIV testing algorithms in several countries in sub-Saharan Africa revealed the inconsistent performances of the testing algorithms at multiple testing sites, some with unacceptable false-positive results [15]. Rapid diagnostic tests are end user-friendly, easy to operate, affordable, but also present some challenges such as errors in performing the test and result interpretation [16]. Additionally, subjective interpretation of the test result can lead to incorrect diagnosis compared to other HIV supplementary assays [17, 18], even among experienced users [19]. In some cases, some testers may be color blind or short-sighted and may not correctly interpret the test outcome [20], or user errors, misinterpretation, suboptimal testing strategies and conditions, poor practices, and clerical errors may further contribute to false diagnosis [21, 22]. In our study, the false positive and negative rates of the rapid test were higher in the field than the NRL; and when compared with HIV multiplex assay. During the study implementation, it was observed that trained and experienced field staff have transitioned out of the program. This might have contributed to the observed performance of the field testing results. Though this was not part of the scope of the study, however, staff attrition has been documented to affect the quality of clinical services, including laboratory services in Nigeria [23, 24].

Emphasis on workforce training, quality assurance, supervisory support, and strengthening laboratory system capacity is crucial for HIV field-based rapid tests [25]. This evaluation suggested the need for comprehensive HIV serology refresher training and supportive supervision for the field staff, while ensuring quality assurance mechanisms are optimized. The need for an accurate, reliable, and timely testing strategy is critical as Nigeria strives toward achieving epidemic control. WHO recommends that all newly diagnosed clients be re-tested to verify their HIV status prior to anti-retroviral therapy (ART) initiation using the same testing algorithm as the initial test [2]. As part of the country's quality management system for HIV serology testing, Nigeria is implementing re-testing for verification of HIV positives before ART initiations [26]. With this national guideline, all HIV positives from the HTS settings are referred to the laboratories for re-testing in line with the adopted country's strategy. This is a significant step as that will reduce the risk of putting about 1.8% of individuals with false positive field results to ART and about 3.4% of false negatives missed, according to our study.

The participation of testing sites in continuous quality improvement, quality assurance, and adherence to the nationally validated HIV testing algorithm is crucial in reducing HIV misdiagnosis, both the initial diagnosis and the re-testing for verification [27]. Evidence shows that country adherence to WHO recommendations for HTS, re-testing for verification, and

appropriate use of the national testing algorithm is vital to achieving accurate HIV diagnosis [28]. A previous study shows that re-testing for verification prior to ART initiation is instrumental to the reduction of HIV misdiagnosis and aids the quality assurance program for HIV testing services [29]. Re-testing also helps to avert significant HIV treatment costs and limit the treatment services to only the confirmed HIV positives in the population [30]. Additionally, the use of only a well-validated HIV rapid testing algorithm should be emphasized as Nigeria approaches HIV epidemic control.

Our study adopted WHO-recommended Phase 2 point of service testing of whole blood for the prospective component of this evaluation. This is the preferred approach for evaluating the performance of an existing algorithm in the field [9]. However, for our retrospective component of the evaluation, data were abstracted from patient records at the facilities in lieu of WHO-recommended laboratory-based evaluation of previously characterized and stored sera. Our methodology provides the flexibility of using existing test results performed under routine field settings without incurring the costs of archiving stored sera. Furthermore, WHO recommended a three reactive test algorithm, especially in regions with HIV prevalence of <5% [2, 14]. This evaluation was informed by the observations from the 2018 NAIIS survey before the WHO recommendation. Therefore, the evaluation focused on validating the performance of the algorithm used for the NAIIS survey. Despite the satisfactory performance of this algorithm, it is recommended that the Federal Ministry of Health consider adopting the recent WHO recommendations to enhance the quality of HIV testing services in the country.

In conclusion, this evaluation confirms that the Nigeria Serial HIV rapid testing algorithm (Determine, Unigold, and Stat-Pak) performed as expected across the target settings. However, there is an urgent need for the Federal Ministry of Health to coordinate comprehensive HIV serology refresher training for the HTS staff in the field and continued implementation of the multilayered quality assurance and continuous quality improvement processes across all HIV testing sites.

## Supporting information

**S1 Table. Summary of the national serial HIV rapid testing algorithm compared with Geenius supplementary assay.**
(DOCX)

## Acknowledgments

The Evaluation Working Group includes Obinna Nnadozie, Ayodele Fagbemi, Grace Bassey, Bello Segun, and Jerry Gwamna. The authors thank all participants that consented to the study. In addition, the authors extend their gratitude to Ado G. Abubakar, Mary Okoli, Chimaoge C. Achugbu, Andrew N. Thomas, Mudiaga K. Esiekpe, Abubakar Iliyasu Bichi, Tamunonengiyeofori Israel, Chinwe N. Ugwu, and Erasogie Evbuomwan for their technical assistance on the HIV multiplex assay. The authors also extend their gratitude to Ndidi Agala for coordinating the sample receipt, accessioning, and storage at the biorepository of the NRL.

## Author Contributions

**Conceptualization:** Nnaemeka C. Iriemenam, Chikwe Ihekweazu, Manhattan Charurat, Mahesh Swaminathan, Bharat S. Parekh, Alash'le Abimiku, McPaul I. Okoye.

**Data curation:** Nnaemeka C. Iriemenam, Olumide O. Okunoye, Akipu Ehoche, Babatunde Ezra, Ernest L. Yufenyuy.

**Formal analysis:** Nnaemeka C. Iriemenam, Matthias A. Alagi, Akipu Ehoche, Babatunde Ezra, Oluwasanmi Adedokun, Brianna Lindsay, Kristen A. Stafford, Ernest L. Yufenyuy.

**Funding acquisition:** Nnaemeka C. Iriemenam, Matthias A. Alagi, Chidozie Meribe, Mukhtar L. Ahmed, Oluwasanmi Adedokun, Dalhatu Ibrahim, Mahesh Swaminathan, Bharat S. Parekh, McPaul I. Okoye.

**Investigation:** Nnaemeka C. Iriemenam, Augustine Mpamugo, Olumide O. Okunoye, Edewede Onokevbagbe, Orji O. Bassey, Jelpe Tapdiyel, Christine Obioha, Ibrahim Baffa Sule, Oluwasanmi Adedokun, Ernest L. Yufenyuy, Bharat S. Parekh, Sylvia Adebajo, Alash'le Abimiku, McPaul I. Okoye.

**Methodology:** Nnaemeka C. Iriemenam, Augustine Mpamugo, Olumide O. Okunoye, Edewede Onokevbagbe, Orji O. Bassey, Jelpe Tapdiyel, Chidozie Meribe, Gabriel Ikwulono, Rose Aguolu, Gregory Ashefor, Charles Nzelu, Christine Obioha, Ibrahim Baffa Sule, Nwando Mba, Alash'le Abimiku, McPaul I. Okoye.

**Project administration:** Nnaemeka C. Iriemenam, Augustine Mpamugo, Akudo Ikpeazu, Olumide O. Okunoye, Edewede Onokevbagbe, Orji O. Bassey, Jelpe Tapdiyel, Matthias A. Alagi, Chidozie Meribe, Mukhtar L. Ahmed, Gabriel Ikwulono, Rose Aguolu, Gregory Ashefor, Charles Nzelu, Christine Obioha, Ibrahim Baffa Sule, Oluwasanmi Adedokun, Nwando Mba, Manhattan Charurat, Dalhatu Ibrahim, Mahesh Swaminathan, Bharat S. Parekh, Sylvia Adebajo, Alash'le Abimiku, McPaul I. Okoye.

**Resources:** Nnaemeka C. Iriemenam, Augustine Mpamugo, Akudo Ikpeazu, Oluwasanmi Adedokun, Nwando Mba, Chikwe Ihekweazu, Manhattan Charurat, Mahesh Swaminathan, Ernest L. Yufenyuy, Sylvia Adebajo, Alash'le Abimiku, McPaul I. Okoye.

**Software:** Akipu Ehoche, Babatunde Ezra, Brianna Lindsay, Kristen A. Stafford.

**Supervision:** Nnaemeka C. Iriemenam, Augustine Mpamugo, Akudo Ikpeazu, Mukhtar L. Ahmed, Gabriel Ikwulono, Rose Aguolu, Gregory Ashefor, Charles Nzelu, Chikwe Ihekweazu, Mahesh Swaminathan, Bharat S. Parekh, Sylvia Adebajo, Alash'le Abimiku, McPaul I. Okoye.

**Validation:** Nnaemeka C. Iriemenam, Augustine Mpamugo, Olumide O. Okunoye, Edewede Onokevbagbe, Ernest L. Yufenyuy, McPaul I. Okoye.

**Visualization:** Nnaemeka C. Iriemenam, Olumide O. Okunoye, Orji O. Bassey.

**Writing – original draft:** Nnaemeka C. Iriemenam.

**Writing – review & editing:** Nnaemeka C. Iriemenam, Augustine Mpamugo, Akudo Ikpeazu, Olumide O. Okunoye, Edewede Onokevbagbe, Orji O. Bassey, Jelpe Tapdiyel, Matthias A. Alagi, Chidozie Meribe, Mukhtar L. Ahmed, Gabriel Ikwulono, Rose Aguolu, Gregory Ashefor, Charles Nzelu, Akipu Ehoche, Babatunde Ezra, Christine Obioha, Ibrahim Baffa Sule, Oluwasanmi Adedokun, Nwando Mba, Chikwe Ihekweazu, Manhattan Charurat, Brianna Lindsay, Kristen A. Stafford, Dalhatu Ibrahim, Mahesh Swaminathan, Ernest L. Yufenyuy, Bharat S. Parekh, Sylvia Adebajo, Alash'le Abimiku, McPaul I. Okoye.

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
