## [Decision Letter · Decision Letter 0]

19 Jul 2022

PGPH-D-22-00701

Evaluation of the Nigeria National HIV Rapid Testing Algorithm

Dear Dr. Iriemenam,

Thank you for submitting your manuscript to PLOS Global Public Health. After careful consideration, we feel that it has merit but does not fully meet PLOS Global Public Health’s publication criteria as it currently stands. Therefore, we invite you to submit a revised version of the manuscript that addresses the points raised during the review process, in particular in positioning the testing algorithm in the context of global standards and recommendations and removing metrics such as PPV that require representative data in terms of disease prevalence to interpret appropriately.

We look forward to receiving your revised manuscript.

Kind regards,

Hannah Hogan Leslie, PhD

Academic Editor

Journal Requirements:

a. Please clarify all sources of funding (financial or material support) for your study. List the grants (with grant number) or organizations (with url) that supported your study, including funding received from your institution. 

b. State the initials, alongside each funding source, of each author to receive each grant.

c. State what role the funders took in the study. If the funders had no role in your study, please state: “The funders had no role in study design, data collection and analysis, decision to publish, or preparation of the manuscript.”

2. We do not publish any copyright or trademark symbols that usually accompany proprietary names, eg (R), (C), or TM  (e.g. next to drug or reagent names). Please remove all instances of trademark/copyright symbols throughout the text, including TM on pages .

Additional Editor Comments (if provided):

Reviewers' comments:

Reviewer's Responses to Questions

**Comments to the Author**

1. Does this manuscript meet PLOS Global Public Health’s publication criteria? Is the manuscript technically sound, and do the data support the conclusions? The manuscript must describe methodologically and ethically rigorous research with conclusions that are appropriately drawn based on the data presented.

Reviewer #1: Yes

Reviewer #2: Partly

2. Has the statistical analysis been performed appropriately and rigorously?

Reviewer #1: Yes

Reviewer #2: No

3. Have the authors made all data underlying the findings in their manuscript fully available (please refer to the Data Availability Statement at the start of the manuscript PDF file)?

Reviewer #1: Yes

Reviewer #2: Yes

4. Is the manuscript presented in an intelligible fashion and written in standard English?

Reviewer #1: Yes

Reviewer #2: Yes

5. Review Comments to the Author

Reviewer #1: This is a well written article. The investigators have done an excellent job of explaining the performance evaluation of Nigeria's national HIV rapid testing algorithm.

It was good to note that Nigeria's HIV rapid testing algorithm was performing as per expectations.

Differences in performance were noted between on-field by regular testers compared to a controlled lab environment by trained staff; the authors have provided logical reasoning for why this difference was observed and attributed it to the fact that the lab staff underwent re-orientation, re-training, and competency assessment. Those with less than 100% scores in practical evaluations were dropped from testing. The authors also provide recommendations to improve the efficiency and performance of the current algorithm. However, the manuscript cannot be accepted without a few revisions. Please find these revisions below.

• "Based on the WHO guidelines for appropriate evaluations of HIV testing technologies in Africa [10], consecutive sampling was done until 1,078 HIV positive and 1,817 HIV negative specimens were collected from all the zones". Authors need to explain the sample size calculation and justify the sample size.

• Figure 1- correction needed: Unigold T2 non-reactive test result is shown as N=338; correct it to 308.

• Under material and methods, the first paragraph incorrectly generalizes the number of states per zone and number of sites per State selected for retrospective analysis to prospective analysis.

"Secondly, a prospective analysis of samples collected from HIV testing sites across the six geopolitical zones with varied HIV prevalence. Two States per geopolitical zone and two sites per State with an annual minimum test capacity of 1,000 HIV rapid tests per year were purposively selected for the evaluation." – this holds for retrospective analysis only. In contrast, one State per zone and three sites per State were selected in prospective analysis. The authors could provide some clarity regarding this information.

• Retrospective study methods: Authors need to either make a reference to figure 1 under the methods section or should explain the algorithm. Authors should mention what variables were abstracted.

• Prospective study methods: Authors need to either refer to figure 1 under the methods section or should explain the algorithm. For example, it is unclear that T2 is applied to only those testing reactive to T1 unless we look at the algorithm (figure 2). Authors need to clarify it in methods to avoid confusion.

•We recommend rewriting some methods sections in the prospective study section to ensure clarity.

• Table 5- comparison of the National HIV Rapid Testing Algorithm performed at the Field and NRL with Genius supplementary essay- authors have provided PPV, agreement rate, and kappa statistics but not the data points. There is no way to validate this calculation. The authors have not mentioned the total number of samples re-tested with the Genius supplementary assay.

Reviewer #2: This study looked at the performance of the HIV testing algorithm in Nigeria.

Major comments

1. The relevance and relatedness of the approaches used to evaluate the algorithm in Nigeria should be reviewed in light of validation protocols that exist to do the same. If there are discrepancies, explaining why this was the case and for what purposes, advantages and disadvantages. This document should be one prioritized: https://www.who.int/publications/i/item/9789240039162. The reference (no10) provided is quite old (2002).

2. A major issue with this study and manuscript is that it focuses on a two-test algorithm. Since 2019, WHO has recommended countries implement a three-test algorithm, regardless of HIV prevalence rate. Presumably Nigeria is also looking to do so. Reflecting on these data in that context and how a three-test strategy may improve the performance data observed would be significantly important. Further, the algorithm presented was not aligned with WHO guidelines (from 2015) – a tie-breaker test has never been recommended, as it is presented here. Understanding the relevance then of the current algorithm presented and whether/how it can be generalized within and beyond Nigeria will be important.

3. Further, the patient selection in the prospective arm was purposeful rather than consecutive. Analyzing the data using PPV and NPV then is generally not possible since a purposeful selection could be biased and is not representative of existing prevalence rates (ie. ~37% with the purposeful vs 1-5% of actual prevalence in Nigeria). This should be removed and more discussion included. Further, there was a complete absence of data and discussion on the sensitivity and specificity of the algorithm in total – this is critical to understand the implications of the data and how it can be utilized in different prevalence settings (which can be calculated within a model).

4. The variable prevalence rates for the country presented are quite confusing and unclear understanding which is correct, how they are different, why they differ, etc. More explanation and rationale here would be important. Additionally, the use of low, medium and high categories within the country seems different known considerations (ie. >5% is deemed high prevalence). Further, in Table 2 though not shown, the calculated prevalence rates were quite high in the high volume settings (higher than the suggested prevalence rates in the background). This should be further explained and likely also included in the table.

5. It is unclear what consent and ethical approval was sought for the retrospective study given data patient was reviewed and may have been captured.

6. In the abstract, ‘changes’ are suggested as a rationale for the work, but were never elaborated.

7. The multiplex assay used should be further described. As is, the work and testing cannot be replicated. It would be helpful to understand if this is a commercial or in-house assay, the protocol used, what type of test, etc. A brief description should be included in addition to the reference provided.

8. The first paragraph of the introduction should be rewritten, it is unclear, especially to non-HIV readers (which could be the case as this is a public health journal, not an HIV one).

9. The use of ‘testing stream’ should be reconsidered. Instead perhaps some standard language such as ‘testing site’ or ‘testing entry point’ or ‘testing location’ may be more explanatory.

10. In Table 2, there could be a few improvements. For example:

a. It isn’t clear what ‘standalone’ means and to what location that refers.

b. It also isn’t clear what ‘ward’ means as it appears different from inpatient. Which ward?

c. There isn’t a TB setting included in this table. Is there no testing in those sites? Please explain.

11. In Table 4, it seems irrelevant to include and mention separately the performance in STI clinics given the sample size. This likely should be included in the ‘Others’ category, especially since it is likely that one or all of those five settings has a higher sample size than STI.

12. The discussion suggests that refresher trainings will support improved field performance; however, there is no evidence and it will likely take a more comprehensive approach. This should be elaborated upon. Further, there was no data presented suggesting that staff attrition may have affected the performance in the field and thus should be removed unless evidence of impact and association can be provided.

13. The discussion introduces a strategy to refer all HIV positives to the laboratory for retesting. This is likely be to concerning given it will cause delays in ART initiation. This should be reflected upon more carefully.

Minor comments

1. On page 6 in the introduction, I believe it should be called ‘three consecutive test algorithm’ not ‘strategies’.

2. On page 12, it would be helpful to elaborate on what pediatric services means with regards to testing entry point. Is it HIV pediatric services, nutrition wards, etc.

6. PLOS authors have the option to publish the peer review history of their article (what does this mean?). If published, this will include your full peer review and any attached files.

**Do you want your identity to be public for this peer review?** For information about this choice, including consent withdrawal, please see our Privacy Policy.

Reviewer #1: No

Reviewer #2: No

---

## [Decision Letter · Decision Letter 1]

14 Sep 2022

PGPH-D-22-00701R1

Evaluation of the Nigeria National HIV Rapid Testing Algorithm

Dear Dr. Iriemenam,

Thank you for submitting your manuscript to PLOS Global Public Health. After careful consideration, we feel that it has merit but does not fully meet PLOS Global Public Health’s publication criteria as it currently stands. I have had the opportunity to share the revision with the reviewer who raised more substantial points on prior review, and we concur that the revision does not adequately address these points in the body of the manuscript; most substantial responses are only in the response to reviewer. I have identified below the most salient issues to address if you consider submitting a revised version. 

We look forward to receiving your revised manuscript.

Kind regards,

Hannah Hogan Leslie, PhD

Academic Editor

Journal Requirements:

Additional Editor Comments (if provided):

1) Clarity in matching the purpose of each element of the study with the reference assay used, calculations performed, and conclusions drawn. The manuscript would benefit from a clear and explicit description of how each element of the study was designed to address the study question, including the choice of quantity (e.g., PPV, NPV, kappa, etc) selected and the threshold applied for interpretation. It is evident that the results of the NAIIS prompted a re-examination of the national testing strategy, which this study then undertakes. It needs to be more clearly laid out that the retrospective assessment was undertaken to consider whether the high rates of discordance in NAIIS were also present in routine testing and whether discordance varied by location, state of the epidemic, or type of testing. The standards applied in terms of acceptable / unacceptable discordance would be helpful to spell out to aid in interpretation.

The prospective evaluation requires similar description and greater clarity in the current analysis. It appears to have two purposes: 1) compare the testing algorithm when undertaken by routine testing services with the same algorithm when conducted in the national lab to see if discrepancy rate is notably higher in the field, and 2) compare algorithm performance against 2 gold standard assays. The method of sampling individuals within each selected site needs to be laid out - it is described as purposive, but based on the response to reviewer, it is only the site selection that was purposive. How were participating individuals selected - convenience in inviting everyone seeking testing on a given date range? The method of selection should be spelled out and the generalizability of participants relative to the target population of everyone being tested for HIV in Nigeria should be considered. The statistical tests used for each comparison and the standards applied should be spelled out; for the second comparison, the reader is not given details on why the Geenius assay was repeated only on positive or discordant tests while the multiplex assay was conducted on all samples. The use of both and the calculation of a range of performance statistics for each introduces confusion. The inclusion of false positive rate comparing the Geenius assay to the NRL and field testing only based on results of samples initially tested as positive or discordant is not appropriate. The use of Kappa and McNemar statistics is not well justified; kappa is used to quantify reliability and does not seem necessary when comparing a test standard to a gold standard. Overall, reducing the number of quantities calculated to ensure that those presented are clear, well justified, and interpreted in reference to a desired standard would greatly enhance the readability of the manuscript. It is also important to address the sample size calculations, which are justified based on a state-level analysis despite the fact that the main analysis is at the national level. Comparisons within each of the 6 sampled areas may be included as supplemental information, or the enrollment of individuals not required to achieve precision at a national level should be justified. 

2) Differences between Nigeria's national approaches and WHO guidelines, including in the use of a tie breaker test and in recommending test confirmation prior to ART initiation - these issues should be made explicit and discussed in greater depth in the discussion section.

3) Implications of the findings for NAIIS - although a stated motivation of the paper is the high discrepancy in testing during the NAIIS, this point is not returned to in the discussion. It is reassuring that the discrepancy rate is much lower in routine practice than it was in the survey, but this merits at least a mention of implications for the reliability of such an important population-based survey.

Reviewers' comments:

Reviewer's Responses to Questions

**Comments to the Author**

1. If the authors have adequately addressed your comments raised in a previous round of review and you feel that this manuscript is now acceptable for publication, you may indicate that here to bypass the “Comments to the Author” section, enter your conflict of interest statement in the “Confidential to Editor” section, and submit your "Accept" recommendation.

Reviewer #2: (No Response)

2. Does this manuscript meet PLOS Global Public Health’s publication criteria? Is the manuscript technically sound, and do the data support the conclusions? The manuscript must describe methodologically and ethically rigorous research with conclusions that are appropriately drawn based on the data presented.

Reviewer #2: Partly

3. Has the statistical analysis been performed appropriately and rigorously?

Reviewer #2: No

4. Have the authors made all data underlying the findings in their manuscript fully available (please refer to the Data Availability Statement at the start of the manuscript PDF file)?

Reviewer #2: Yes

5. Is the manuscript presented in an intelligible fashion and written in standard English?

Reviewer #2: Yes

6. Review Comments to the Author

Reviewer #2: (No Response)

7. PLOS authors have the option to publish the peer review history of their article (what does this mean?). If published, this will include your full peer review and any attached files.

**Do you want your identity to be public for this peer review?** For information about this choice, including consent withdrawal, please see our Privacy Policy.

Reviewer #2: No

---

## [Editor Report · Decision Letter 2]

28 Sep 2022

PGPH-D-22-00701R2

Evaluation of the Nigeria National HIV Rapid Testing Algorithm

Dear Dr. Iriemenam,

Thank you for submitting your manuscript to PLOS Global Public Health. After careful consideration, we feel that it has merit but does not fully meet PLOS Global Public Health’s publication criteria as it currently stands. Therefore, we invite you to submit a revised version of the manuscript that addresses the points raised during the review process.

I appreciate the rapid response to the previous round of reviews and the changes made. As in the first revision, I believe it would be helpful if the extensive comments in the response to review were more fully reflected in the actual changes in the manuscript, particularly in making clear the purpose and interpretation of the tests conducted.

PLOS Global Public Health is a deliberately cross-disciplinary journal with a wide readership; providing clarity on the purpose of the specific laboratory tests used is necessary to contextualize this work for that readership in a way that perhaps it would not be for a laboratory or even HIV-specific journal. In terms of statistical tests performed, certainly kappa statistics can be calculated, but the article does not interpret the added value they provide over and above the sensitivity/specificity PPV/NPV statistics laid out in testing guidelines since as WHO's. In addition, the issue raised regarding the sample size - calculated for sub-regional analysis not included in this work - remains unaddressed.

We look forward to receiving your revised manuscript.

Kind regards,

Hannah Hogan Leslie, PhD

Academic Editor
---

## [Editor Report · Decision Letter 3]

10 Oct 2022

Evaluation of the Nigeria National HIV Rapid Testing Algorithm

PGPH-D-22-00701R3

Dear Dr Iriemenam,

We are pleased to inform you that your manuscript 'Evaluation of the Nigeria National HIV Rapid Testing Algorithm' has been provisionally accepted for publication in PLOS Global Public Health.

Best regards,

Hannah Hogan Leslie, PhD

Academic Editor